**Data Availability Statement:** All relevant data are within the paper and its Supporting Information files.

**Funding:** The author(s) received no specific funding for this work.

# Prophylactic effects of probiotics or synbiotics on postoperative ileus after gastrointestinal cancer surgery: A meta-analysis of randomized controlled trials

Gang Tang[1], Wang Huang[1], Jie Tao[2], Zhengqiang Wei🄳[1]*

**1** Department of Gastrointestinal Surgery, The First Affiliated Hospital of Chongqing Medical University, Chongqing, China, **2** Department of Clinical Medicine, Chongqing Medical University, Chongqing, China

\* 1980900547@qq.com

## Abstract

### Background

Postoperative ileus is a major problem following gastrointestinal cancers surgery, several randomized controlled trials have been conducted investigating the use of probiotics or synbiotics to reduce postoperative ileus, but their findings are controversial.

### Objective

We conducted a meta-analysis to determine the effect of probiotics or synbiotics on early postoperative recovery of gastrointestinal function in patients with gastrointestinal cancer.

### Methods

The Embase, Cochrane Library, PubMed, and Web of Science databases were comprehensively searched for randomized controlled trials (RCTs) that evaluated the effects of probiotics or synbiotics on postoperative recovery of gastrointestinal function as of April 27, 2021. Outcomes included the time to first flatus, time to first defecation, days to first solid diet, days to first fluid diet, length of postoperative hospital stay, incidence of abdominal distension and incidence of postoperative ileus. The results were reported as the mean difference (MD) and relative risk (RR) with 95% confidence intervals (CI).

### Results

A total of 21 RCTs, involving 1776 participants, were included. Compared with the control group, probiotic and synbiotic supplementation resulted in a shorter first flatus (MD, -0.53 days), first defecation (MD, -0.78 days), first solid diet (MD, -0.25 days), first fluid diet (MD, -0.29 days) and postoperative hospital stay (MD, -1.43 days). Furthermore, Probiotic and synbiotic supplementation reduced the incidence of abdominal distension (RR, 0.62) and incidence of postoperative ileus (RR, 0.47).

**Competing interests:** The authors have declared that no competing interests exist.

## Conclusion

Perioperative supplementation of probiotics or synbiotics can effectively promote the recovery of gastrointestinal function after gastrointestinal cancer surgery.

## Introduction

Gastrointestinal cancers account for about 25% of new cancer cases worldwide and cause more than 35% of cancer-related deaths [1]. Surgery is an essential treatment for gastrointestinal cancer. Postoperative ileus is an inevitable and most common complication of gastrointestinal surgery, with up to 30% of patients suffering from postoperative ileus [2–4]. Postoperative ileus refers to the delayed recovery of gastrointestinal function after surgery, with clinical manifestations of abdominal distension, abdominal pain, vomiting, and delayed defecation of exhaust, leading to prolonged hospital stay and increased morbidity [4–7]. Postoperative ileus is a significant financial burden for patients, adding more than 1,000,000,000 dollars in additional medical costs annually in the United States [8]. Although a number of strategies have been explored for the prevention of postoperative ileus, such as gum chewing, intravenous lidocaine, and preoperative activities, their efficacy remains controversial [5].

Probiotics are living microorganisms that are beneficial to the human body when supplemented in appropriate amounts [9]. Prebiotics are substances, such as inulin and fructooligosaccharides that promote beneficial gut microbe growth [10]. Probiotics combined with prebiotics are called synbiotics [9]. Historically, probiotics and synbiotics have been widely used in the adjuvant treatment of gastrointestinal diseases [11]. In recent years, a large number of studies have found that probiotics and synbiotics can reduce the risk of infection complications after abdominal surgery [12]. In addition, probiotics and synbiotics could also promote gastrointestinal motility [13]. Probiotics and synbiotics are inexpensive, readily available, and safe [14]. Based on these findings, probiotics and synbiotics may be potential strategies to promote recovery of gastrointestinal function after gastrointestinal cancer surgery and to reduce the incidence of postoperative ileus. However, clinical studies have shown conflicting results [15, 16]. Therefore, it is extremely important to establish strong evidence to determine whether perioperative probiotics or synbiotics can prevent postoperative ileus.

Hence, we systematically collected evidence from current randomized controlled trails (RCTs) and performed a meta-analysis to determine the effect of probiotics or synbiotics on early postoperative recovery of gastrointestinal function in patients with gastrointestinal cancer.

## Materials and methods

The meta-analysis is reported based on the Preferred Reporting Items for Systematic Reviews and Meta-Analyses (PRISMA) statement [17] (see S1 Checklist, PRISMA checklist, which contains PRISMA 2009 checklist).

### Search strategy

Systematic literature searches were conducted on Web of Science, Cochrane Library, Embase, and PubMed databases with no filters until April 27, 2021. The search terms were: (synbiotics OR prebiotic OR probiotics OR probiotic OR prebiotics OR synbiotic) AND (operation OR

surgery) AND (cancer OR neoplasm OR carcinoma OR tumour) (S1 Table). Additionally, the reference lists of related reviews were also searched to reduce omissions.

## Study selection

Studies that met the following criteria were included: (1) study design: RCTs, (2) participants: gastrointestinal cancer patients undergoing surgery, (3) intervention: intervention with probiotics or synbiotics, (4) comparison: the control group received the standard treatment or a placebo, and (5) the outcomes included any of the following: time to first flatus, time to first defecation, postoperative ileus, days to first solid diet, abdominal distension, days to first fluid diet, and length of postoperative hospital stay. Duplicate studies, reviews, abstracts, non-randomized trails, animal studies, letters, and case reports were excluded.

## Data extraction

The first author, gender, year, primary disease, sample size, type of surgery, type of study, age, treated days, intervention, control group data, and outcomes were extracted from each study. If the essential data could not be obtained from the article, the corresponding author was contacted to try to obtain the missing data.

## Quality assessment

Risk of bias for eligible studies was assessed by the ROB-2 tool available in the Cochrane Handbook, including the following domains: (1) Randomization process, (2) Deviations from intended interventions, (3) Missing outcome data, (4) Measurement of the outcome, (5) Selection of the reported result, and (6) Overall. Literature retrieval, selection of article, data extraction, and risk of bias assessment were performed independently by two authors (Gang Tang and Jie Tao). If there was a disagreement between the authors, it was discussed and resolved with a third author (Wang Huang).

## Statistical analysis

For continuous data, the mean differences (MD) with 95% confidence intervals (CIs) were calculated. Relative risks (RRs) were calculated for dichotomous variable data [18]. The $I^2$ statistic was used to assess the magnitude of heterogeneity between studies; The random effect model was used in all quantitative analyses, and the fixed effect model was selected only when heterogeneity was low [19]. For result robustness, the one-study exclusion test was used to investigate the influence of each study on the total effect size. Subgroup analysis was performed by intervention type (probiotics or synbiotics). Egger's test was performed using Stata 12.0 (Stata Corp., College Station, TX, USA) to assess potential publication bias. In addition, funnel plots were used when the number of included studies > 10. All statistical analyses were performed using Review 5.3 (The Nordic Cochrane Centre, The Cochrane Collaboration 2014; Copenhagen, Denmark). P <0.05 was considered significant.

## GRADE assessment

To grade the quality of evidence, a GRADE assessment was performed through GRADEpro online tools (https://gradepro.org/). GRADE assessed the evidence as four levels: very low, low, medium, and high. The two researchers (Gang Tang and Jie Tao) independently assess the certainty of the evidence, and if there was dispute, they would discuss and resolve it.

## Results

### Literature retrieval

Our search strategy yielded 1,992 records and 463 duplicates were removed. 1479 of the results were excluded after reading the headings and abstracts, and the remaining 50 records were evaluated for the full text. Finally, 21 eligible studies [16, 20–39] were included. The flow chart of literature retrieval is shown in Fig 1.

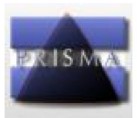 **PRISMA 2009 Flow Diagram**

```
Identification
```

Records identified through database searching (n = 1990)

Additional records identified through other sources (n = 2)

```
Screening
```

Records after duplicates removed (n = 1529)

Records screened (n = 50)

Records excluded (n = 1479)

```
Eligibility
```

Full-text articles assessed for eligibility (n = 21)

Full-text articles excluded, with reasons (n = 29)
1. Not RCT (n= 2)
2. No probiotic or synbiotic interventions were received (n= 5)
3. Not cancer patients were received (n= 12)
4. Probiotics or synbiotic are used in combination with other drugs (n= 2)
4. Target outcomes are not reported (n= 8)

```
Included
```

Studies included in quantitative synthesis (meta-analysis) (n = 21)

**Fig 1. Flow chart of literature search and screening.**

## Study characteristics

Between 2005 and 2020, 21 studies were published with 1776 total participants (875 in the intervention group and 901 in the control group). Twelve studies [16, 20, 22, 25, 29, 30, 32, 35–39] used only probiotics, and nine [21, 23, 24, 26–28, 31, 33, 34] used synbiotics. The indications for surgery were colorectal cancer, gastric cancer, liver cancer, gallbladder cancer, esophageal cancer and periampullary cancer. The characteristics of eligible studies are detailed in S2 Table.

## Quality assessment

Ten of the studies [16, 22, 23, 27–29, 31, 32, 34, 36] conducted an appropriate randomization process. Deviations from intended interventions were evaluated as a low bias risk in six studies [16, 20, 22, 31, 34, 36]. Missing outcome data, measurement of the outcome, and selection of the reported result in all studies were assessed as a low bias risk (Fig 2). The overall risk of 10 studies [16, 22, 23, 27–29, 31, 32, 34, 36] was assessed as low risk of bias.

## Meta-analysis

**Time to first flatus.** Eight RCTs [20, 24, 28, 32, 33, 37–39] (617 patients) reported on time to first flatus. Probiotics or synbiotics supplementation was associated with a significant

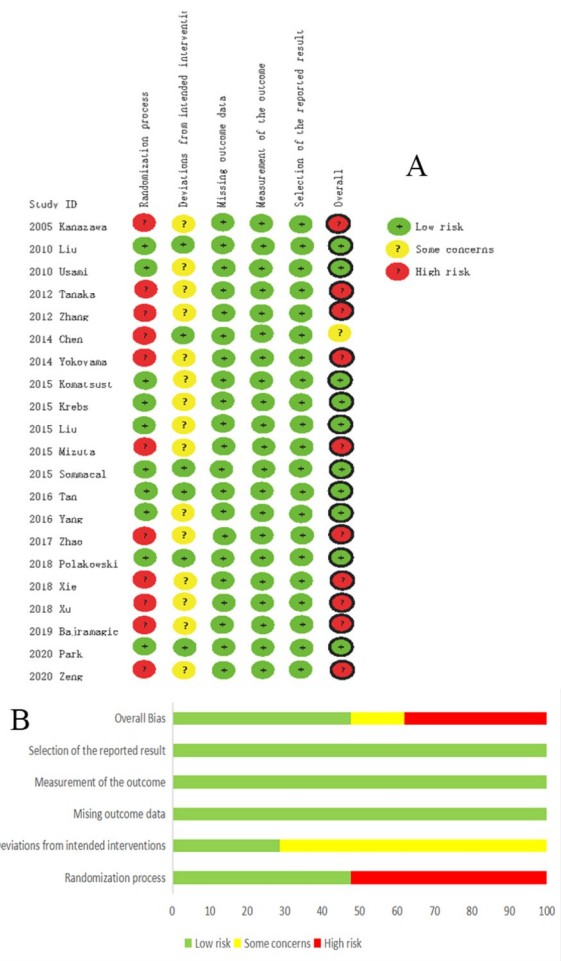

**Fig 2. Risk of bias for each included study.** (A), risk of bias summary. (B), risk of bias graph.

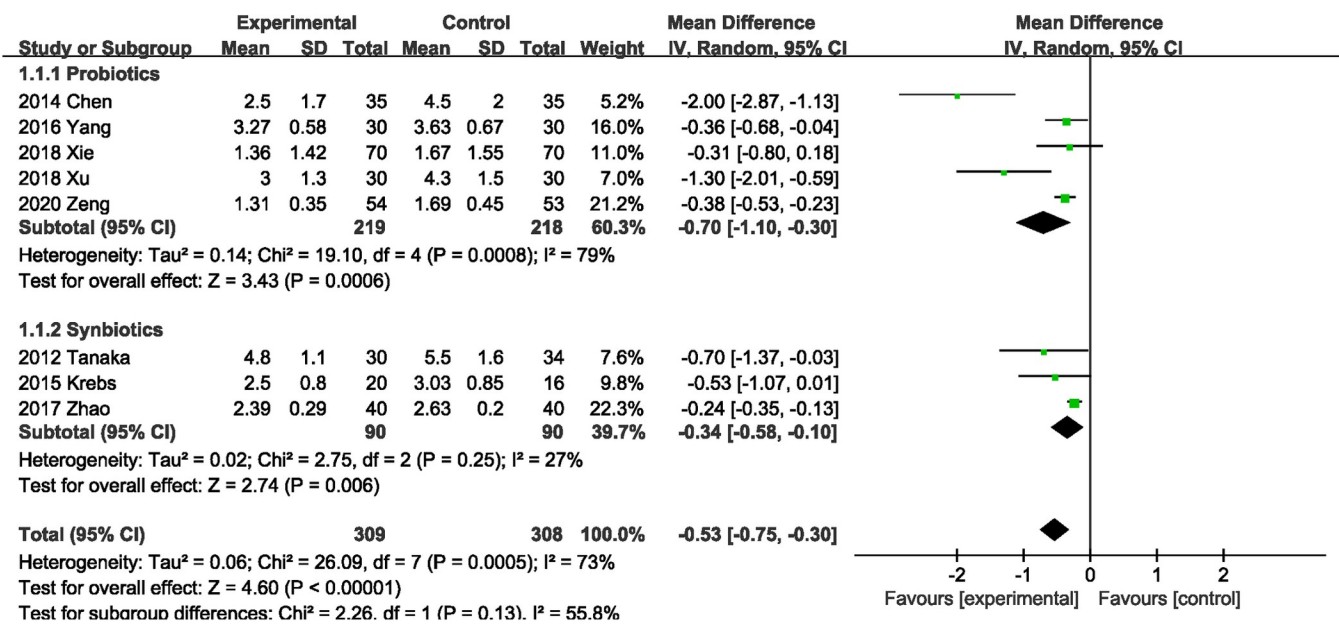

**Fig 3. Effect of probiotics or synbiotics supplementation on time to first flatus.**

reduction in time to first flatus (MD, -0.53 days; 95% CI, -0.75, -0.30; P < 0.00001) (Fig 3), with significant heterogeneity ($I^2$ = 73%, P = 0.0005). The results of subgroup analysis showed that both probiotics (MD, -0.70 days; 95% CI, -1.10, -0.30; P = 0.0006) alone and synbiotics (MD, -0.34 days; 95% CI, -0.58, -0.10; P = 0.006) supplementation were associated with shorter first exhaust time.

**Time to first defecation.** Seven studies [20, 22, 24, 28, 29, 32, 39] measured time to first defecation as an outcome. Compared with the control group, probiotics or synbiotics significantly reduced the time to first defecation, with significant heterogeneity (MD, -0.78 days; 95% CI, -1.27, -0.28; P = 0.002; $I^2$ = 86%, P < 0.00001) (Fig 4). Subgroup analysis indicated that this

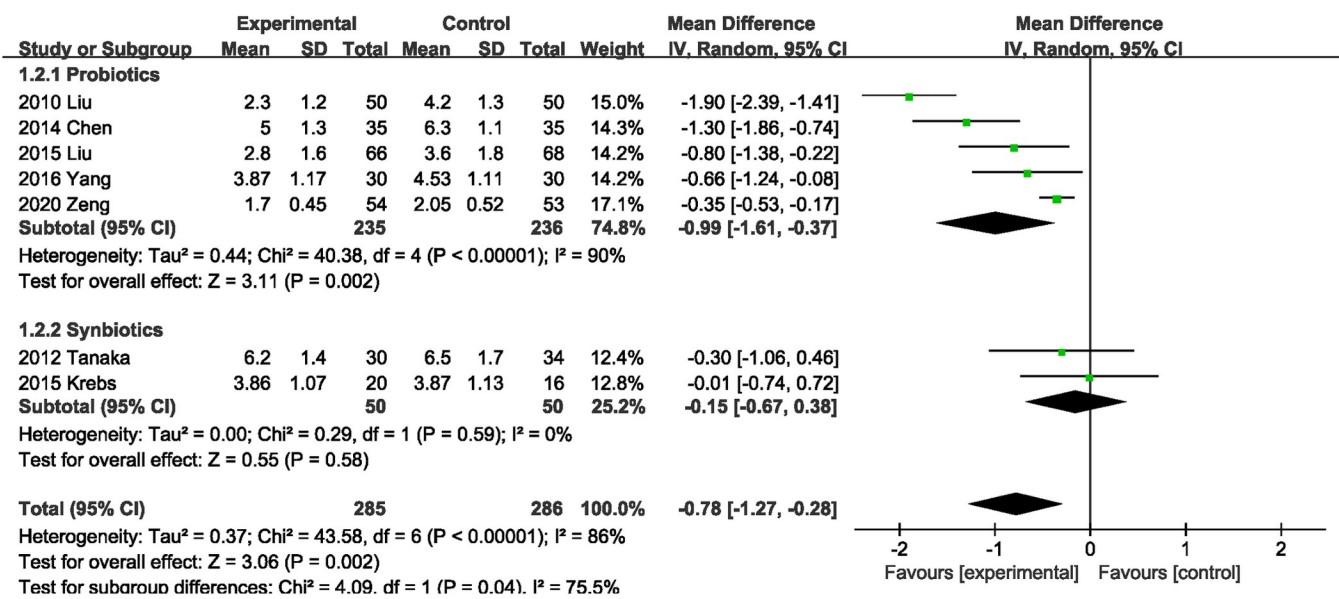

**Fig 4. Effect of probiotics or synbiotics supplementation on time to first defecation.**

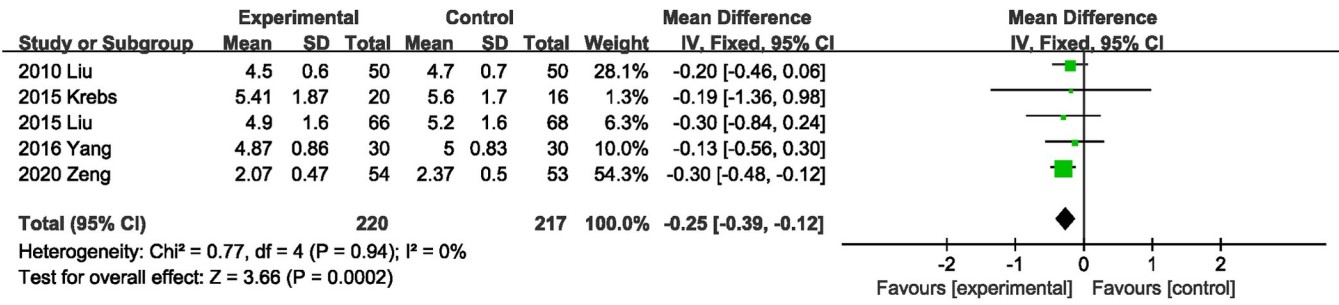

**Fig 5. Effect of probiotics or synbiotics supplementation on days to first solid diet.**

benefit was observed only in the subgroup supplemented with probiotics alone (MD, -0.99 days; 95% CI, -1.61, -0.37).

**Days to first solid diet.** Five studies [22, 28, 29, 32, 39] reported data for days to first solid diet, pooled results showed that probiotics or synbiotics supplementation significantly shortened the days to first solid diet (MD, -0.25 days; 95% CI, -0.39, -0.12; P = 0.0002) (Fig 5). In addition, no significant heterogeneity was shown between RCTs ($I^2$ = 0%, P = 0.94).

**Days to first fluid diet.** Three RCTs [22, 29, 32] mentioned days to first fluid diet. Probiotics or synbiotics significantly shortened the days to first fluid diet (MD, -0.29 days; 95% CI, -0.47, -0.11; P = 0.001) (Fig 6), and no significant heterogeneity was observed between the three studies ($I^2$ = 0%, P = 0.83).

**Length of postoperative hospital stay.** Twelve RCTs [16, 21–23, 26, 29–33, 37] with a total of 440 participants were in the probiotics or synbiotics group and 440 in the control. The combined result favored probiotics or synbiotics supplementation, with a MD of 1.43 days reduction (MD, -1.43 days; 95% CI, -2.29, -0.58; P = 0.001; $I^2$ = 67%; Fig 7). Subgroup analysis showed that both probiotics (MD, -1.06 days; 95% CI, -2.05, -0.07; P = 0.04) and synbiotics (MD, -2.34 days; 95% CI, -4.29, -0.39; P = 0.02) supplementation reduced length of postoperative hospital stay.

**Postoperative ileus.** Of the 21 eligible RCTs, four studies [25, 27, 35, 36] (559 participants) reported findings on postoperative ileus, the combined total effect size showed that supplementation with probiotics or synbiotics significantly reduced the incidence of postoperative ileus (RR, 0.47; 95% CI, 0.24, 0.91, P = 0.02; $I^2$ = 9%, P = 0.35) (Fig 8).

**Abdominal distension.** Five RCTs [20, 22, 29, 32, 33] presented data on incidence of abdominal distension. Supplementation with probiotics or synbiotics was associated with a significant reduction in the incidence of postoperative abdominal distension (RR, 0.62; 95% CI, 0.47, 0.81; P = 0.0004) (Fig 9), with low heterogeneity ($I^2$ = 0%, P = 0.97).

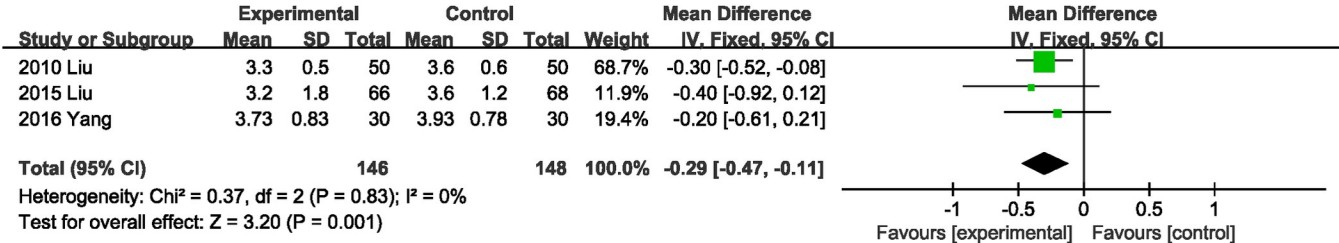

**Fig 6. Effect of probiotics or synbiotics supplementation on days to first fluid diet.**

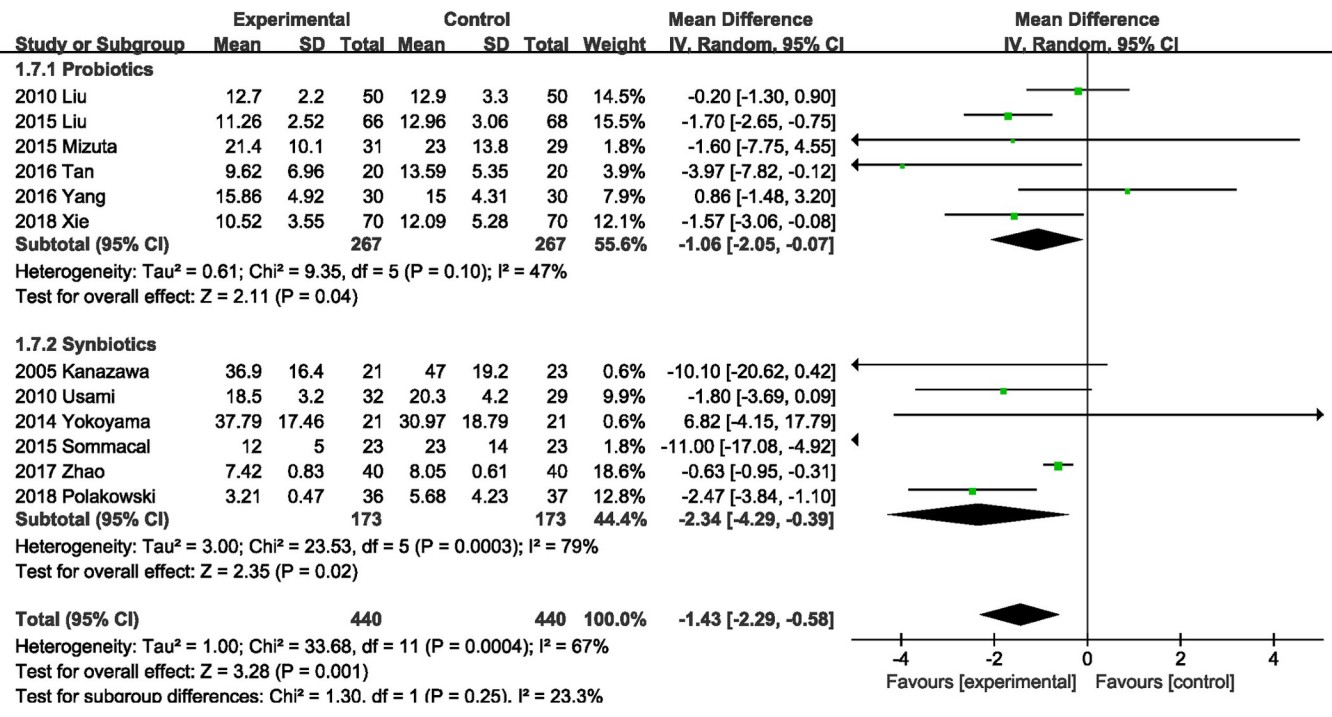

**Fig 7. Effect of probiotics or synbiotics supplementation on length of postoperative hospital stay.**

## Sensitivity analysis

The results of the sensitivity analysis indicated that excluding any one study did not affect the total effect size of the time to first flatus, time to first defecation, days to first solid diet, length of postoperative hospital stay and incidence of abdominal distension. The overall effect size for the days to first fluid diet changed when the study by Liu et al. [22] (MD, -0.28 days; 95% CI, -0.60, 0.04; P = 0.09) was excluded. The overall effect size of the incidence of postoperative ileus was influenced by the study of Bajramagic et al. [35] (RR, 0.66; 95% CI, 0.32, 1.37, P = 0.26).

## Publication bias

Egger's test results did not show potential publication bias of the time to first flatus (P = 0.214), time to first defecation (P = 0.754), days to first solid diet (P = 0.609), days to first fluid diet

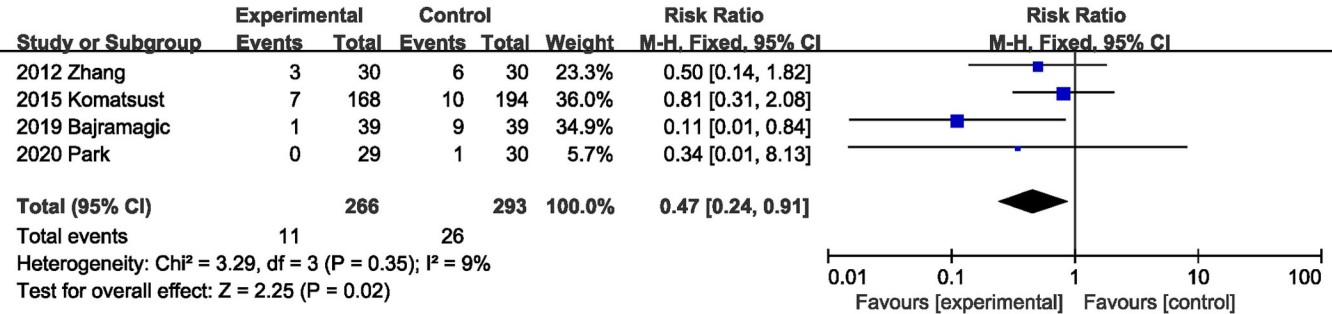

**Fig 8. Effect of probiotics or synbiotics supplementation on the incidence of postoperative ileus.**

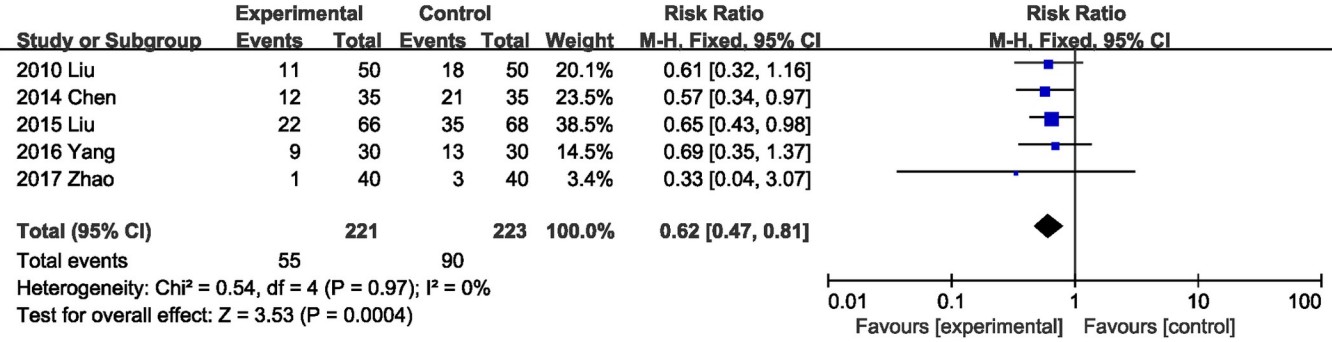

**Fig 9. Effect of probiotics or synbiotics supplementation on the incidence of postoperative abdominal distension.**

(P = 0.991), length of postoperative hospital stay (P = 0.970), incidence of abdominal distension (P = 0.530) and incidence of postoperative ileus (P = 0.265). Visual inspection of the funnel plot (length of postoperative hospital stay) identified basically symmetric (Fig 10).

## GRADE analysis

We evaluated the quality of evidence in this study (Fig 11). A part of the evidence (the time to first flatus, days to first fluid diet, incidence of abdominal distension and incidence of

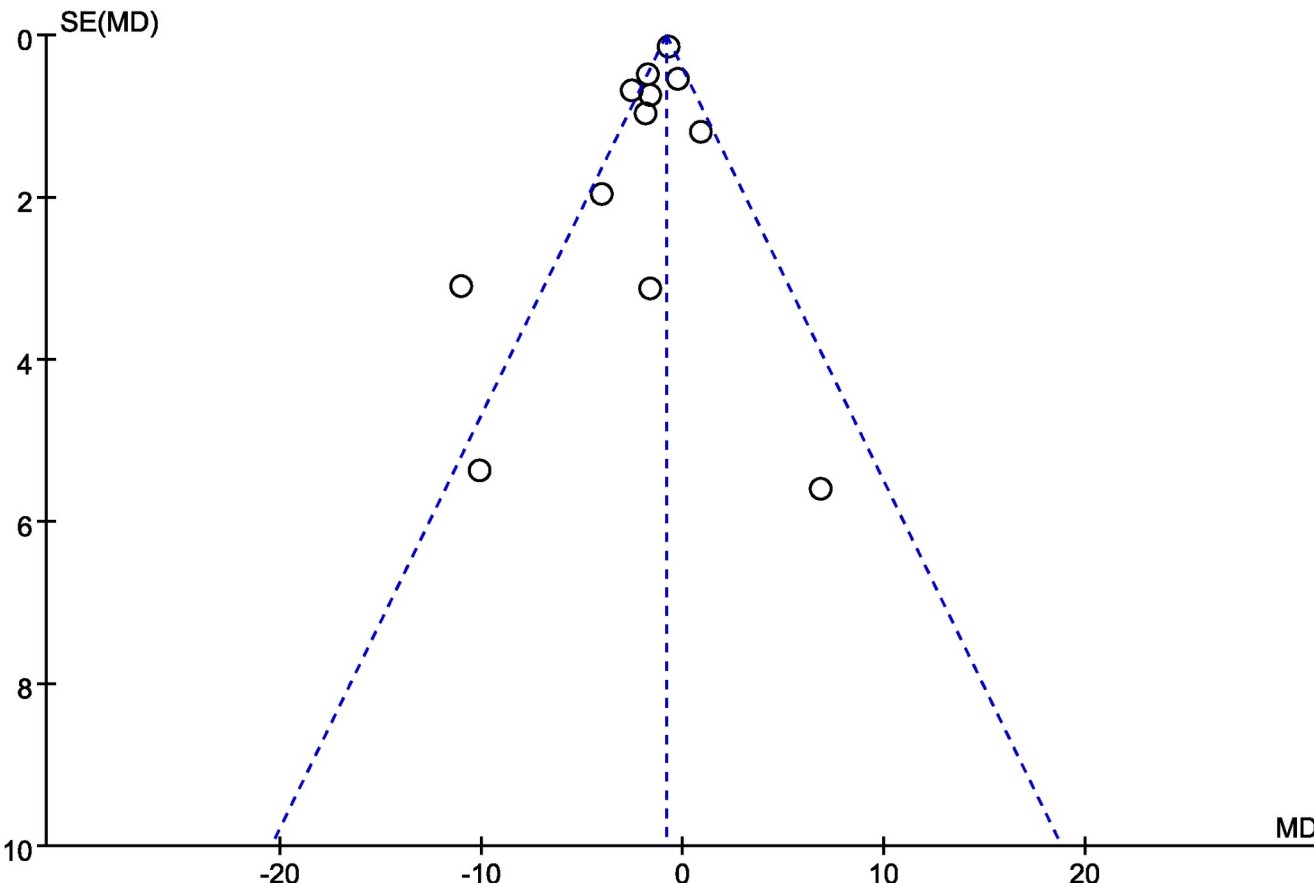

**Fig 10. Funnel plot of effect of probiotics or synbiotics supplementation on the length of postoperative hospital stay.**

| № of studies | Study design | Risk of bias | Inconsistency | Indirectness | Imprecision | Other considerations | Probiotics or synbiotics | placebo | Relative (95% CI) | Absolute (95% CI) | Certainty | Importance |
|---|---|---|---|---|---|---|---|---|---|---|---|---|
| | | | | | | | | | | | Certainty | Importance |
| **The time to first flatus** | | | | | | | | | | | | |
| 8 | randomised trials | not serious | serious [a] | not serious | not serious | none | 309 | 308 | - | MD **0.53 lower** (0.75 lower to 0.3 lower) | ⊕⊕⊕◯ Moderate | CRITICAL |
| **The time to first defecation** | | | | | | | | | | | | |
| 7 | randomised trials | not serious | very serious [b] | not serious | not serious | none | 285 | 286 | - | MD **0.78 lower** (1.27 lower to 0.28 lower) | ⊕⊕◯◯ Low | CRITICAL |
| **Days to first solid diet** | | | | | | | | | | | | |
| 5 | randomised trials | not serious | not serious | not serious | not serious | none | 220 | 217 | - | MD **0.25 lower** (0.39 lower to 0.12 lower) | ⊕⊕⊕⊕ High | CRITICAL |
| **Days to first fluid diet** | | | | | | | | | | | | |
| 3 | randomised trials | not serious | not serious | not serious | serious [c] | none | 146 | 148 | - | MD **0.29 lower** (0.47 lower to 0.11 lower) | ⊕⊕⊕◯ Moderate | CRITICAL |
| **Incidence of postoperative ileus** | | | | | | | | | | | | |
| 4 | randomised trials | not serious | not serious | not serious | serious [c] | none | 11/266 (4.1%) | 26/293 (8.9%) | RR **0.47** (0.24 to 0.91) | **47 fewer per 1,000** (from 67 fewer to 8 fewer) | ⊕⊕⊕◯ Moderate | IMPORTANT |
| **Incidence of abdominal distension** | | | | | | | | | | | | |
| 5 | randomised trials | not serious | not serious | serious [d] | not serious | none | 55/221 (24.9%) | 90/223 (40.4%) | RR **0.62** (0.47 to 0.81) | **153 fewer per 1,000** (from 214 fewer to 77 fewer) | ⊕⊕⊕◯ Moderate | IMPORTANT |
| **Length of postoperative hospital stay** | | | | | | | | | | | | |
| 12 | randomised trials | not serious | very serious [e] | serious [f] | not serious | none | 440 | 440 | - | MD **1.43 lower** (2.29 lower to 0.58 lower) | ⊕◯◯◯ Very low | NOT IMPORTANT |

**CI:** confidence interval; **MD:** mean difference; **RR:** risk ratio

**Explanations**

a. Downgraded because the I^2 value was 73%
b. Downgraded because the I^2 value was 86%
c. Sensitivity analysis showed that the robustness of the results was affected by individual study
d. As the incidence of abdominal distension is a indirect indicator for postoperative ileus. Hence, downgraded for indirectness
e. Downgraded because the I^2 value was 67%
f. As length of postoperative hospital stay is a indirect indicator for recovery of intestinal function. Hence, downgraded for indirectness

**Fig 11. Grade evidence synthesis and summary of findings.**

postoperative ileus) was in a medium level, one (length of postoperative hospital stay) was very low, one (the time to first defecation) was low, one (days to first solid diet) was high.

## Discussion

Postoperative gastrointestinal function, as the core part of the accelerated recovery of patients with gastrointestinal cancer undergoing surgery, has important clinical significance and has been paid close attention by surgeons [40]. To our knowledge, this is the first meta-analysis to evaluate the effect of probiotics or synbiotics on gastrointestinal function recovery after gastro-intestinal cancer surgery. Evidence from this meta-analysis was based on 21 RCTs with 1776 participants. The results showed that peri-operative probiotics or synbiotics supplementation signifcantly reduced the time to first flatus, time to first defecation, days to first solid diet, days to first fluid diet and length of postoperative hospital stay. The time to first flatus and time to first defecation are the key to evaluate gastrointestinal dysfunction and postoperative ileus. They are generally considered to be the relief of postoperative ileus, and are also important indicators to evaluate the efficacy of intervention methods [4]. The results of subgroup analysis showed that either probiotics alone or synbiotics alone could shorten the time to first exhaust and first defecation. In addition, probiotics or prebiotics could also reduce the incidence of postoperative abdominal distension and postoperative ileus. This study has important clinical significance because our meta-analysis provides clear evidence that probiotics or synbiotics could promote gastrointestinal recovery normality after surgery for gastrointestinal cancer. Hence, probiotics or synbiotics are potential strategies that clinicians should consider in the prevention of postoperative ileus.

The mechanism of postoperative ileus is not clear and may involve the interaction of many factors [2], inhibition of gastrointestinal motility caused by surgical overstimulation of the sympathetic nerve may be the most important factor [41]. In addition, substance P and nitric oxide secreted by the enteric nervous system also prolong the duration of postoperative ileus.

Furthermore, surgery stimulates the inflammatory cascade, releasing a large number of inflammatory mediators, such as interleukin-6, interleukin-1, monocytechemoattractantprotein-1 and cell adhesion molecule-1, which damage intestinal muscles and further inhibit the recovery of gastrointestinal function [2, 41]. Some drugs have also been associated with increased the risk of ileus after surgery [2]. Probiotics or synbiotics are an alternative therapy widely used in cancer patients to prevent postoperative infection, relieve symptoms and improve quality of life, with beneficial effects in a variety of gastrointestinal diseases having been demonstrated [42]. Peri-operative supplementation with probiotics or synbiotics could modulate local and systemic immune homeostasis, reduce inflammatory responses, and reduce concentrations of pro-inflammatory factors, tumor necrosis factor-α, interleukin-6, C-reactive protein, and nitric oxide which could aggravate postoperative ileus by ameliorating operationally induced intestinal flora dysregulation [42–46]. In addition, Schmitter et al. found that probiotics significantly reduced the release of interleukin-6, interleukin-8, and prostaglanin E2 from monocytes compared with placebo [47]. Studies have shown that dendritic cells in the gastrointestinal tract can interact with intestinal nerve cells and intestinal microorganisms. Probiotics or synbiotics may stimulate nerve cells to promote gastrointestinal function recovery by regulating intestinal microorganisms [42].

Several excluded clinical studies have also supported the beneficial effects of probiotics or synbiotics on postoperative ileus. A non-RCT study by Aisu et al. [48] showed that perioperative probiotics supplementation significantly reduced the time to first exhaust and first feeding. Kotzampassi et al. [49] found that a capsule containing four probiotics significantly shortened the time to first defecation in patients undergoing colorectal surgery, compared with a placebo. In addition, Xu et al. [50] demonstrated that early use of synbitin after colon cancer surgery can improve immune function, reduce inflammatory response, and promote gastrointestinal function recovery.

This study has several strengths. First, only RCTs were included in our meta-analysis in order to synthesize the strongest evidence. Second, this study conducted a comprehensive literature search to reduce bias. Furthermore, we used advanced statistical methods to find no potential publication bias. Finally, we confirmed the robustness of our results (including time to first exhaust, time to first defecate, days to first fluid diet, incidence of abdominal distension and length of hospital stay) through sensitivity analysis.

Our meta-analysis also had several limitations. First, several studies with small sample sizes were included. Second, some outcome measures (incidence of postoperative ileus and incidence of postoperative abdominal distension) were quantitatively synthesized based on a small number of studies. Third, Significant heterogeneity was observed in our study, which may be related to significant differences in type of surgery (radical colorectomy, liver resection, esophagectomy, colorectal cancer resection, gastrectomy and pancreatoduodenectomy), duration of probiotics or synbiotics supplementation (from 3 days to 28 days), species of probiotics or synbiotics and dose of probiotics or synbiotics. Future research should explore the specific species of probiotics or synbiotics with the greatest benefit for gastrointestinal function recovery, as well as the most appropriate course and dose of probiotics or synbiotics supplementation. Finally, this study only included patients with gastrointestinal cancer who underwent elective surgery, so our findings may not be generalizable to patients undergoing emergency surgery.

## Conclusions

In conclusion, our study showed that perioperative supplementation of probiotics or synbiotics can effectively promote the recovery of gastrointestinal function after gastrointestinal cancer surgery, including shorting the time to first flatus, time to first defecation, days to first solid

diet, days to first fluid diet and length of postoperative hospital stay, and reducing the incidence of postoperative abdominal distention and postoperative ileus. But these conclusions need to be treated with caution, given some limitations that cannot be ignored. High-quality, large-sample RCTs are necessary to confirm the benefit of probiotics or synbiotics supplementation for gastrointestinal function recovery after gastrointestinal cancer surgery.

## Supporting information

**S1 Checklist. PRISMA checklist.**
(DOC)

**S1 Table. Electronic search strategy.**
(DOC)

**S2 Table. Characteristics of 21 eligible studies.** CFU: colony forming units; C: Control group; DB: Double blind; I: Intervention group; GOS: galacto-oligosaccharides; PD: pancreatoduodenectomy; N: not available; RCT: randomized controlled trial; SC: standard care; TF: time to first flatus; TD: time to first defecation; LOP: Length of postoperative hospital stay; PI: Postoperative ileus; DS: Days to first solid diet; AB; abdominal distension; DF: days to first fluid die.
(DOC)

## Author Contributions

**Conceptualization:** Gang Tang, Wang Huang, Jie Tao, Zhengqiang Wei.

**Data curation:** Jie Tao.

**Formal analysis:** Gang Tang, Wang Huang, Jie Tao, Zhengqiang Wei.

**Investigation:** Gang Tang, Zhengqiang Wei.

**Methodology:** Gang Tang, Zhengqiang Wei.

**Software:** Gang Tang, Wang Huang, Jie Tao, Zhengqiang Wei.

**Supervision:** Zhengqiang Wei.

**Validation:** Zhengqiang Wei.

**Visualization:** Zhengqiang Wei.

**Writing – original draft:** Gang Tang, Wang Huang, Jie Tao.

**Writing – review & editing:** Gang Tang, Wang Huang, Zhengqiang Wei.

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
