## [Decision Letter · Decision Letter 0]

8 Sep 2021

PONE-D-21-20250Prophylactic Effects of Probiotics or Synbiotics on Postoperative Ileus After Gastrointestinal Cancer Surgery: A Meta-analysis of Randomized Controlled TrialsPLOS ONE

Dear Dr. Wei,

Thank you for submitting your manuscript to PLOS ONE. After careful consideration, we feel that it has merit but does not fully meet PLOS ONE’s publication criteria as it currently stands. Therefore, we invite you to submit a revised version of the manuscript that addresses the points raised during the review process.

We look forward to receiving your revised manuscript.

Kind regards,

Laura Pasin

Academic Editor

PLOS ONE

Journal Requirements:

2. We note that this manuscript is a systematic review or meta-analysis; our author guidelines therefore require that you use PRISMA guidance to help improve reporting quality of this type of study. Please upload copies of the completed PRISMA checklist as Supporting Information with a file name “PRISMA checklist

Reviewers' comments:

Reviewer's Responses to Questions

**Comments to the Author**

1. Is the manuscript technically sound, and do the data support the conclusions?

Reviewer #1: Yes

Reviewer #2: Partly

2. Has the statistical analysis been performed appropriately and rigorously? 

Reviewer #1: Yes

Reviewer #2: No

3. Have the authors made all data underlying the findings in their manuscript fully available?

Reviewer #1: Yes

Reviewer #2: Yes

4. Is the manuscript presented in an intelligible fashion and written in standard English?

Reviewer #1: Yes

Reviewer #2: Yes

5. Review Comments to the Author

Reviewer #1: Thank you for your work.

The manuscript is interesting and well written.

As underlined by authors in the discussion, the two main limitations are the heterogeneity of included studies ( very different types of surgery) and the small sample size of included studies.

I would like to ask to the authors if they performed a trial sequential analysis, to better evaluate the results of their study.

Thank you

Reviewer #2: GENERAL COMMENT

Thanks for the opportunity to review this article. The authors conducted a systematic review and meta-analysis of randomized controlled trials investigating the effect of probiotics or synbiotics on early postoperative recovery of gastrointestinal function in patients with gastrointestinal cancer. They found that perioperative supplementation of probiotics or synbiotics can effectively promote the recovery of gastrointestinal function after gastrointestinal cancer surgery. The research question is clinically relevant and original. However, I would have the following major comments:

- I would suggest the authors to perform an extensive English language editing of the manuscript.

- Further efforts should be carried out to explore the reasons for the moderate-high heterogeneity detected among the studies, as the authors appropriately pointed out in the limitations section. This would significantly strengthen the authors’ findings.

- The Risk of Bias tool the authors employed is not the most recent one. The recommended tool for assessing risk of bias of RCTs is the RoB 2 (https://sites.google.com/site/riskofbiastool/welcome/rob-2-0-tool/current-version-of-rob-2?authuser=0). Please provide the risk of bias assessment by using this tool.

- A GRADE assessment is very important to better inform the reader about the overall certainty of evidence. Please perform a GRADE assessment. Online tools are available for this, e.g., https://gradepro.org/.

MINOR COMMENTS

Line 77: “Our meta-analysis was conducted based on”

The meta-analysis is reported based on PRISMA Checklist, rather than conducted based on it. Please modify accordingly.

Line 81: “Additionally, the reference lists of related reviews were also searched”

It seems to me that this sentence should be placed afterwards or removed, considering the last sentence of the paragraph.

Lines 110-111: “the random-effects model was selected when I^2 was >50 %”

It is not recommended to choose the model based on the I^2 value. A random-effects model should be applied regardless of the I^2, unless the authors anticipate there to be very small heterogeneity. Otherwise, sensitivity analyses with different I^2 thresholds should be performed.

Line 113: “Egger's test”

Along with the Egger’s test, should there be a significant number of included studies (> 10), funnel plots should be provided as well.

Lines 145-146: “and there was significant heterogeneity (I^2 = 55.8%, P = 0.13) between subgroups. In addition, heterogeneity was significantly reduced in the synbiotics subgroup (I^2 = 27%, P = 0.25)”

In my opinion, the authors should report that the p-value allows to conclude that there was no subgroup difference. The difference in heterogeneity between the groups cannot really be compared and stated as significant.

Line 152: “with significant heterogeneity”

The authors should provide the I^2 and the p-value relative to the I^2 test, like for the time to first flatus outcome. The same should be considered for Lines 178-179.

6. PLOS authors have the option to publish the peer review history of their article (what does this mean?). If published, this will include your full peer review and any attached files.

Reviewer #1: No

Reviewer #2: No

---

## [Author Response · Author response to Decision Letter 0]

26 Sep 2021

Replies to the Reviewer 

Thank you very much for taking time to review our manuscript. We have carefully read your comments and suggestions, which have been of great help. We have revised the manuscript according to the comments and responded point-by-point to the comments, as listed below. Additionally, we have highlighted all revised text.

Reviewer #1: Thank you for your work.

The manuscript is interesting and well written.

As underlined by authors in the discussion, the two main limitations are the heterogeneity of included studies ( very different types of surgery) and the small sample size of included studies.

I would like to ask to the authors if they performed a trial sequential analysis, to better evaluate the results of their study.

Thank you

Answer: Thanks again to the reviewer, we did not performe a trial sequential analysis in our meta-analysis. This is the first meta-analysis on this topic, so we did not consider using a trial sequential analysis when making our study protocol. 

Reviewer #2: GENERAL COMMENT

Thanks for the opportunity to review this article. The authors conducted a systematic review and meta-analysis of randomized controlled trials investigating the effect of probiotics or synbiotics on early postoperative recovery of gastrointestinal function in patients with gastrointestinal cancer. They found that perioperative supplementation of probiotics or synbiotics can effectively promote the recovery of gastrointestinal function after gastrointestinal cancer surgery. The research question is clinically relevant and original. However, I would have the following major comments:

- I would suggest the authors to perform an extensive English language editing of the manuscript.

Answer: With the help of the Taylor & Francis Editing Services, we perform an extensive English language editing of the manuscript. The major revisions are highlighted in the manuscript, and the minor revisions made to fix grammatical errors and improve expressions do not affect the results.

- Further efforts should be carried out to explore the reasons for the moderate-high heterogeneity detected among the studies, as the authors appropriately pointed out in the limitations section. This would significantly strengthen the authors’ findings.

Answer: Thanks for the reviewer's suggestion. Due to the significant heterogeneity, we are prepared to conduct a subgroup analysis to explore the source of heterogeneity. As we mentioned in the discussion, different surgical methods, differences dose and species of probiotics or synbiotics, and different courses of treatment may be the source of heterogeneity. However, due to the limited number of studies, we were unable to conduct subgroup analysis of these factors. In the future, we will conduct more detailed analyses with more studies.

In the discussion section of the manuscript : “Significant heterogeneity was observed in our study, which may be related to significant differences in type of surgery (radical colorectomy, liver resection, esophagectomy, colorectal cancer resection, gastrectomy and pancreatoduodenectomy), duration of probiotics or synbiotics supplementation (from 3 days to 28 days), species of probiotics or synbiotics and dose of probiotics or synbiotics. Future research should explore the specific species of probiotics or synbiotics with the greatest benefit for gastrointestinal function recovery, as well as the most appropriate course and dose of probiotics or synbiotics supplementation.”

- The Risk of Bias tool the authors employed is not the most recent one. The recommended tool for assessing risk of bias of RCTs is the RoB 2 (https://sites.google.com/site/riskofbiastool/welcome/rob-2-0-tool/current-version-of-rob-2?authuser=0). Please provide the risk of bias assessment by using this tool.

Answer: As suggested by reviewers, we have used the latest tools for bias evaluation (figure 2).

“Risk of bias for eligible studies was assessed by the ROB-2 tool available in the Cochrane Handbook, including the following domains: (1) Randomization process, (2) Deviations from intended interventions, (3) Missing outcome data, (4) Measurement of the outcome, (5) Selection of the reported result, and (6) Overall.”

“Ten of the studies [22, 23, 27-29, 31-33, 35, 37] conducted an appropriate randomization process. Deviations from intended interventions were evaluated as a low bias risk in six studies [20, 22, 31, 32, 35, 37].Missing outcome data, measurement of the outcome, and selection of the reported result in all studies were assessed as a low bias risk (Fig 2). The overall risk of 10 studies [22, 23, 27-29, 31-33, 35, 37] was assessed as low risk of bias.”

- A GRADE assessment is very important to better inform the reader about the overall certainty of evidence. Please perform a GRADE assessment. Online tools are available for this, e.g., https://gradepro.org/.

Answer: Thanks you for your comments, we have performed a GRADE assessment and revised the manuscript. As follow: “ GRADE Assessment

To grade the quality of evidence, a GRADE assessment was performed through GRADEpro online tools (https://gradepro.org/). GRADE assessed the evidence as four levels: very low, low, medium, and high. The two researchers (Gang Tang and Jie Tao) independently assess the certainty of the evidence, and if there was dispute, they would discuss and resolve it.”

“GRADE analysis

We evaluated the quality of evidence in this study (Fig 11). A part of the evidence (the time to first flatus , days to first fluid diet , incidence of abdominal distension and incidence of postoperative ileus) was in a medium level, one (length of postoperative hospital stay) was very low, one (the time to first defecation) was low, one (days to first solid diet) was high”

MINOR COMMENTS

Line 77: “Our meta-analysis was conducted based on”

The meta-analysis is reported based on PRISMA Checklist, rather than conducted based on it. Please modify accordingly.

Answer: Thanks to the reviewers for pointing out these details, we have revised them in the manuscript . 

As follow: “The meta-analysis is reported based on the Preferred Reporting Items for Systematic Reviews and Meta-Analyses (PRISMA) statement”

Line 81: “Additionally, the reference lists of related reviews were also searched”

It seems to me that this sentence should be placed afterwards or removed, considering the last sentence of the paragraph.

Answer: As suggested by the reviewer, we deleted this sentence.

Lines 110-111: “the random-effects model was selected when I^2 was >50 %”

It is not recommended to choose the model based on the I^2 value. A random-effects model should be applied regardless of the I^2, unless the authors anticipate there to be very small heterogeneity. Otherwise, sensitivity analyses with different I^2 thresholds should be performed.

Answer: We revised the manuscript as suggested by the reviewers. As follow: “The random effect model was used in all quantitative analyses, and the fixed effect model was selected only when heterogeneity was low”

Line 113: “Egger's test”

Along with the Egger’s test, should there be a significant number of included studies (> 10), funnel plots should be provided as well.

Answer: We have made the following revisions.

“In addition, funnel plots were used when the number of included studies > 10.”

“Visual inspection of the funnel plot (length of postoperative hospital stay) identified basically symmetric (Fig 10).”

Lines 145-146: “and there was significant heterogeneity (I^2 = 55.8%, P = 0.13) between subgroups. In addition, heterogeneity was significantly reduced in the synbiotics subgroup (I^2 = 27%, P = 0.25)”

In my opinion, the authors should report that the p-value allows to conclude that there was no subgroup difference. The difference in heterogeneity between the groups cannot really be compared and stated as significant.

Answer: We fully agree with the reviewer's point of view, and we have deleted this sentence

Line 152: “with significant heterogeneity”

The authors should provide the I^2 and the p-value relative to the I^2 test, like for the time to first flatus outcome. The same should be considered for Lines 178-179.

Answer: We revised the manuscript based on suggestions from reviewers, and we have provided the I^2 and the p-value relative to the I^2 test.

---

## [Decision Letter · Decision Letter 1]

17 Feb 2022

Prophylactic Effects of Probiotics or Synbiotics on Postoperative Ileus After Gastrointestinal Cancer Surgery: A Meta-analysis of Randomized Controlled Trials

PONE-D-21-20250R1

Dear Dr. wei,

We’re pleased to inform you that your manuscript has been judged scientifically suitable for publication and will be formally accepted for publication once it meets all outstanding technical requirements.

Kind regards,

Laura Pasin

Academic Editor

PLOS ONE

Additional Editor Comments (optional):

Reviewers' comments:

Reviewer's Responses to Questions

**Comments to the Author**

1. If the authors have adequately addressed your comments raised in a previous round of review and you feel that this manuscript is now acceptable for publication, you may indicate that here to bypass the “Comments to the Author” section, enter your conflict of interest statement in the “Confidential to Editor” section, and submit your "Accept" recommendation.

Reviewer #1: All comments have been addressed

Reviewer #3: All comments have been addressed

2. Is the manuscript technically sound, and do the data support the conclusions?

Reviewer #1: (No Response)

Reviewer #3: Yes

3. Has the statistical analysis been performed appropriately and rigorously? 

Reviewer #1: (No Response)

Reviewer #3: Yes

4. Have the authors made all data underlying the findings in their manuscript fully available?

Reviewer #1: (No Response)

Reviewer #3: Yes

5. Is the manuscript presented in an intelligible fashion and written in standard English?

Reviewer #1: (No Response)

Reviewer #3: Yes

6. Review Comments to the Author

Reviewer #1: (No Response)

Reviewer #3: Thank you for allowing me to review this manuscript entitled “Prophylactic Effects of Probiotics or Synbiotics on Postoperative Ileus After Gastrointestinal Cancer Surgery: A Meta-analysis of Randomized Controlled Trials”

I have read with interest the article, which is overall well written, and I believe the findings are clinically significant. The methodology appears to be mostly sound and the conclusions are supported by the data.

A few comments:

I suggest providing funnel plots for all outcomes and performing a visual inspection, as this can be useful to interpret the results of Egger’s test, which is also not recommended in the Cochrane handbook when there are less than 10 studies (I suggest the relevant chapter in the cochrane handbook: https://training.cochrane.org/handbook/archive/v6/chapter-13).

I suggest adding to the limits that the subgroup analysis for the type of surgery, dose and species of probiotics/synbiotics was not possible due to the limited number of studies, as mentioned in the answer to Reviewer 1.

Aside from these aspects which in my opinion could benefit from clarification, I believe the manuscript is overall sound and interesting to read. I congratulate the authors for their interesting research.

7. PLOS authors have the option to publish the peer review history of their article (what does this mean?). If published, this will include your full peer review and any attached files.

Reviewer #1: No

Reviewer #3: No

---

## [Editor Report · Acceptance letter]

21 Feb 2022

PONE-D-21-20250R1 

Prophylactic Effects of Probiotics or Synbiotics on Postoperative Ileus After Gastrointestinal Cancer Surgery: A Meta-analysis of Randomized Controlled Trials 

Dear Dr. Wei:

I'm pleased to inform you that your manuscript has been deemed suitable for publication in PLOS ONE. Congratulations! Your manuscript is now with our production department. 

Kind regards, 

on behalf of

Dr. Laura Pasin 

Academic Editor

PLOS ONE